# Dietary Intake of Adolescents and Alignment with Recommendations for Healthy and Sustainable Diets: Results of the SI.Menu Study

**DOI:** 10.3390/nu16121912

**Published:** 2024-06-17

**Authors:** Rok Poličnik, Hristo Hristov, Živa Lavriša, Jerneja Farkaš, Sonja Smole Možina, Barbara Koroušić Seljak, Urška Blaznik, Matej Gregorič, Igor Pravst

**Affiliations:** 1National Institute of Public Health, Trubarjeva Cesta 2, SI-1000 Ljubljana, Slovenia; jerneja.farkas@sb-ms.si (J.F.); urska.blaznik@nijz.si (U.B.); matej.gregoric@nijz.si (M.G.); 2Nutrition Institute, Koprska Ulica 98, SI-1000 Ljubljana, Slovenia; hristo.hristov@nutris.org (H.H.); ziva.lavrisa@nutris.org (Ž.L.); igor.pravst@nutris.org (I.P.); 3Biotechnical Faculty, University of Ljubljana, Jamnikarjeva Ulica 101, SI-1000 Ljubljana, Slovenia; sonja.smole-mozina@bf.uni-lj.si; 4Faculty of Medicine, University of Ljubljana, Vrazov trg 2, SI-1000 Ljubljana, Slovenia; 5General Hospital Murska Sobota, Ulica dr. Vrbnjaka 6, Rakičan, Sl-9000 Murska Sobota, Slovenia; 6Computer System Department, Jožef Stefan Institute, Jamova Cesta 39, SI-1000 Ljubljana, Slovenia; barbara.korousic@ijs.si; 7VIST-Faculty of Applied Sciences, Gerbičeva Cesta 51A, SI-1000 Ljubljana, Slovenia

**Keywords:** nutrient intake, energy intake, dietary intake, adolescents, 24 h recall, FPQ, dietary survey, Slovenia, sustainable diet

## Abstract

Background: The SI.Menu study offers the latest data on the dietary intake of Slovenian adolescents aged 10 to 17. The purpose of this study is to comprehensively assess their dietary intake (energy and nutrients) and compare their food intakes with dietary recommendations for healthy and sustainable diets. Methods: The cross-sectional epidemiological dietary study SI.Menu (March 2017–April 2018) was conducted on a representative sample of Slovenian adolescents aged 10 to 17 years (*n =* 468) (230 males and 238 females). Data on dietary intake were gathered through two non-consecutive 24 h recalls, in line with the European Food Safety Authority (EFSA) EU Menu methodology. The repeated 24 h Dietary Recall (HDR) and Food Propensity Questionnaire (FPQ) data were combined to determine the usual intakes of nutrients and food groups, using the Multiple Source Method (MSM) program. Results: Adolescents’ diets significantly deviate from dietary recommendations, lacking vegetables, milk and dairy products, nuts and seeds, legumes, and water, while containing excessive meat (especially red meat) and high-sugar foods. This results in insufficient intake of dietary fibre, and nutrients such as vitamin D, folate, and calcium. Conclusions: The dietary intake of Slovenian adolescents does not meet healthy and sustainable diet recommendations. This study provides an important insight into the dietary habits of Slovenian adolescents that could be useful for future public health strategies.

## 1. Introduction

Adolescence is a time of dynamic growth and development, with notable implications for an individual’s future health. This critical phase is nutrition-sensitive, and proper nutrition has many benefits for various physiological systems beyond just growth [1].

Like other developed countries [2,3], Slovenia has been facing unhealthy eating habits and an increase in obesity among children and adolescents in recent decades [4]. It is well known that inappropriate dietary habits and consequently inappropriate dietary intake may lead to the growth of obesity [1] and to other diet-related diseases [5]. 

In terms of health protection, the period of growing into young adulthood represents an important part of an individual’s life, since it is characterized by rebelliousness, growing independence, and a distancing from dietary recommendations. During this period, an increased shift to unhealthy dietary choices was reported [6], which can be a major cause of excessive energy intake. Adolescents are often exposed to the influences of modern social media, which excessively prefer unhealthy dietary choices and a sedentary lifestyle [7,8]. Research shows that unhealthy lifestyles among children and adolescents, coupled with the increase in obesity and metabolic syndrome, leads to nutrient deficiency due to insufficient intake of iron, calcium, zinc, vitamin D, and certain other micronutrients, that are vital for optimal growth and development [5]. 

Similar to other European countries, Slovenia is facing a lack of regular, nationally representative, and sufficiently accurate longitudinal studies that would offer insight into dietary trends and nutritional intake of children and adolescents. Accordingly, Slovenia joined the European Food Safety Authority (EFSA) EU Menu project in 2017, which introduced the standardization of data collection on food consumption, ensuring greater comparability of food consumption data among the EU countries [9,10]. 

Recently, there has been a growing body of scientific literature highlighting the significance of promoting responsibility for sustainable development, reducing carbon footprints, and fostering environmental and natural protection in relation to dietary practices [11,12,13]. The integration of these principles into national dietary guidelines has already started in some parts of Europe, including in the Nordic countries [14] and the Netherlands [15]. Meanwhile, other countries are still in the process of developing their guidelines [16], or data on this are not available in the literature [17]. 

The aim of this article is to present the latest data on the dietary intake of Slovenian adolescents aged 10 to 17 years, collected in 2017/2018, in accordance with the EFSA EU Menu protocol, with a particular focus on the consumption of specific food groups. The primary objective of this study was to examine the energy consumption and dietary intake of macronutrients and micronutrients in adolescents, comparing the findings with the national dietary references [18] and the Dutch healthy and sustainable food-based dietary guidelines (FBDG) [15]. Additionally, we assess the influence of different sociodemographic, anthropometric, and other confounding variables affecting the consumption of selected food groups.

## 2. Materials and Methods

### 2.1. Study Design and Population

The cross-sectional Slovenian National Food Consumption Survey, SI.Menu 2017/2018, was carried out between March 2017 and April 2018. The study design followed the EFSA Guidance on EU Menu Methodology [9] and is described in detail elsewhere [10]. In short, the target population within the SI.Menu project was Slovenian residents aged between 10 and 74 years residing in private households. Individuals living abroad or in institutional households, ill, and people with disabilities were excluded from the study. Subjects were sampled from the Central Population Registry by the Statistical Office of the Republic of Slovenia. A two-stage probability sampling method was used with stratification by cohesion regions (NUTS-2 cohesion region) and degree of urbanization (DEGURBA). In every sampling unit, subjects were selected and adequate representation according to sex and age was assured. A total of 2280 individuals from all 12 statistical regions in Slovenia were selected, of whom 2119 were found eligible. Invited subjects were classified into three age groups—adolescents (10–17 years), adults (18–64 years), and elderlies (65–74 years). The present study only includes adolescent participants, while data for adults and elderlies have been published recently [19]. Participation in the research was voluntary and parents’ consent was requested. With the aim of comparing with the national dietary references for energy and dietary intake of macro- and micronutrients [18], we divided the adolescents into three age groups (10–12; 13–14; and 15–17 years). For the purpose of comparing the consumption of specific food groups with the Dutch healthy and sustainable FBDG [15], we divided the adolescents into two age groups (10–13; and 14–17 years). The study was conducted in accordance with the principles of the Declaration of Helsinki, and it was approved by the Medical Ethics Committee of the Republic of Slovenia under the number KME 0120-337/2016. 

### 2.2. Dietary and Anthropometric Data Assessment

Various tools and protocols were developed to implement the dietary study, including the Food Propensity Questionnaire (FPQ) [10], a validated picture book [20] to estimate the amount of food consumed, the International Physical Activity Questionnaire (IPAQ) [21], and the SI.Menu web-based dietary application. Additionally, sociodemographic and other household size and composition-related data were obtained. The complete study design, protocols, and tools were described in detail elsewhere [10]. Fieldwork was carried out using the open code application SurveyToGo [22] and the SI.Menu application, which is based on the Open Platform for Clinical Nutrition tool (OPEN) [23] and contains national data on the nutritional composition of foods and traditional and frequently used recipes. The SI.Menu application was adapted to support 24 h dietary recalls (24 HDRs) as well as food diaries. It enabled the entry of respondent data (type of meal, place and time of meal), and detailed data on foods and beverages (and food supplements) consumed in the preceding day [10,20]. The data used for assessment of energy and nutrient intakes were obtained by means of a 24 h dietary recall method, carried out on two non-consecutive days, in accordance with the EFSA guidance [9]. The estimates of consumed portion sizes were facilitated by the nationally developed and validated picture book [20], which was supplemented by household kitchen measurements and portions indicated in standard recipes. Fieldwork was conducted with the support of computer-assisted personal interviews [10]. Previously trained researchers who followed the standard EFSA recommended study protocol performed face-to-face interviews and anthropometric measurements, including margining participants’ height and weight. Each interview took around 30 min and was repeated with each respondent within 21 days. In the planning of interviews, an equal distribution of different days of the week (both week days and weekend days) and different seasons (including all four seasons of the year) was considered [24]. Body mass and height were measured using standardized medical equipment and procedures. The cut-off point for overweight and obesity was adjusted by sex/age taking into account all participants’ deviation by at least one standard deviation (SD) of the mean [25,26]. The underweight category included participants with BMIs less than the 5th percentile based on sex and age group affiliation as suggested by Timmerman et al. [27].

### 2.3. Food Categorisation

All foods and beverages reported in 24 HDRs were inserted into OPEN and linked with food composition data. For the purpose of the analyses of dietary intakes, food recipes were disaggregated into the ingredients based on their weight included in the recipes. The daily food consumption was based on the edible part of the food and the portion size quantified as consumed weight. To investigate food sources of nutrients, we designed a food classification scheme based on the categories used for assessing dietary intakes from the FPQ. A total of 2377 different foods and beverages were reported in both 24 HDRs. Foods were initially classified into 101 food categories, of which 78 were covered by the FPQ, and 23 categories were additionally developed to meet the reporting needs and were assessed only habitually using two 24 HDRs [24]. For the purpose of this study, all foods were further regrouped into 20 main categories, and 18 subcategories, according to the modified categorisation system adapted from Haubrock et al. [28] and corresponding to the FBDG categories based on the DGE (Deutsche Gesellschaft für Ernährung) Nutrition Circle [29]. A detailed description of the food categories and subcategories is presented in Appendix A.

### 2.4. Data Analysis

#### 2.4.1. Sample Selection: Under- and Over-Reporting

Of the 722 contactable adolescent individuals, 495 participated in the survey and 485 provided two completed 24 HDRs. After the exclusion of 9 participants with missing anthropometric data and/or incomplete survey responses, a complete dataset was available for 476 study participants (244 males and 232 females), forming the basis for the present study.

Handling under- and over-reporters was previously described and explained [30]. In short, estimated subjects’ energy intakes were assessed using the cut-off points method initially described by Goldberg et al. [31] and further adapted by Black et al. [32]. The method is based on the ratio of reported daily energy intake and estimated basic metabolic rate (BMR) based on the method described by Harris and Benedict [33] and adapted by Roza and Shizgal [34]. The calculated cut-off points for 24 HDRs for under- and over-reporting were 0.41 and 2.46, resulting in the exclusion of 7 adolescent subjects. Additionally, one subject that reported low energy intake in a magnitude of less than 500 kcal/day was excluded from the analysis. Summarizing the aforementioned, a total of 27 individuals (14 males and 13 females) out of those who agreed to participate in the study were excluded from the analysis. Added to the one that refused to participate in the study after signing the consent statement, the overall response rate was 64.8%. For the purpose of this study, subjects were further divided into three age groups (10–12 years: *n =* 194; 13–14 years: *n =* 93; and 15–17 years: *n =* 181); the analysis was conducted separately for male and female subjects (Table 1).

#### 2.4.2. Statistical Analysis

All statistical analyses were performed using STATA (version 17.0; StataCorp LLC, College Station, TX, USA). The Multiple Source Method (MSM) statistical program was used to estimate the usual intake distribution of foods. The repeated 24 HDR and FPQ data were combined to estimate the distribution of usual food groups consumption based on statistical models that take into account a statistical correction for within-subject variation [28]. Day-to-day inter- and intra-individual variations in the intake distribution were modelled using MSM, where age, sex, and BMI were used as covariates. The MSM modelling approach used FPQ data to correct within-individual variation in food and nutrient intake, providing data on usual dietary intake on an individual level [35]. To avoid the problem of a high level of non-consumers in the modelling of usual food intakes for individual food items, MSM was used on the parent food category level (Appendix A). It should be noted that for eggs, data were reported only as 24 HDR intake, because this food group was not investigated using the FPQ. The modelling of usual daily food intakes with MSM was done with transformed FPQ frequencies (i.e., FPQ frequency 1 time/week was converted to 1/7 per day). The same food (sub)groups were subsequently used to estimate also energy and nutrient intakes on the individual level—first on the food subgroup level, and later on upper group level. 

The descriptive data are presented as mean and standard deviation for the continuous variables or as counts and percentages for categorised variables. Usual intake data (food groups, energy, nutrients)—calculated separately for males/females and adolescent age groups—are presented as mean (per day) with standard deviation (SD), and with median/percentiles (5%, 25%, 50%, 75%, and 95%). Nutrient intakes were calculated both in grams per day, and as the percentage of total energy intake (TEI). For energy and proteins, intake is also presented per kg of body weight (BW) (in kJ or grams, respectively). Additionally, we calculated and present the results for the number of true non-consumers and the probability of consumption of selected food groups for both sexes (Appendix A).

TEI and nutrient intakes were compared with previous national data, and those obtained from the relevant literature describing the usual dietary intakes for comparable population and sex groups [18]. Total water intakes were compared with adequate intake in the EFSA’s dietary reference values for water [36]. We calculated the proportion of subjects (%) not meeting sex- and age-group-adapted reference values for daily intakes of energy, macronutrients, dietary fibre, water, and micronutrients as reported in the national dietary references [18]. The same approach was used for food groups, where comparison was done using cut-offs reported by Brink et al., 2019 [15] related to the healthy and sustainable FBDG for the adolescent population in the Netherlands. Where needed, the data referring to weekly consumed amounts (i.e., for fish, meat, and eggs) were converted into daily amounts. To identify the potential determinants of influence (cohesion regions, sex, age groups, BMI, degree of urbanization, IPAQ (self-assessed), household composition, sampling season) explaining variation in consumed daily amounts across different food groups, we used the least absolute shrinkage and selection operator (LASSO) regression method as suggested by Freese et al. [37]. This procedure is recommended as the most appropriate for variable selection in linear mixed effect models. Although it relies upon the linear regression model, it goes beyond this by introducing shrinkage by minimizing the objective function, penalising some coefficients towards zero, and addressing issues like multicollinearity and variable selection. The final selected model consists of a small subset of the predictors—namely, those with nonzero coefficient estimates that affect the response variable. LASSO is a valuable tool for understanding how multiple factors (in our study, sociodemographic, anthropometric, and other confounding variables) influence a single outcome (specific food group). The most suitable LASSO model was selected according to the adaptive method.

## 3. Results

The Slovenian National Food Consumption Survey SI.Menu 2017/18 included 468 adolescent participants (49.1% males: *n =* 230; and 50.9% females: *n =* 238) aged 10 to 17 years. Sample characteristics of participants are described in Table 1. The highest proportion of overweight or obese participants among particular age groups were found in the age group 13 to 14 years (17.2%). Additionally, the highest percentage of underweight participants (6.5%) was also observed in this group. Altogether, 46.2% of all participants were identified as high-intensity physically active based on self-assessment using the International Physical Activity Questionnaire (IPAQ) [21].

Table 2 and Table 3 present the usual daily intakes of food groups among adolescents in Slovenia, by age groups and sex. Study results indicate that males across all age categories have higher usual daily intakes of milk compared to females. The lowest usual mean daily milk intake was observed in those aged 15 to 17 years (105.9 ± 149.6 g/day) (Table 2). 

As expected, adolescents consumed more fruits than vegetables. The vegetable intake in females slightly decreases with age, while in males a slight but not indicative increase in intake of all vegetable categories (fresh and canned) can be detected with age increase. Slovenian adolescents consume legumes below the recommended levels, with an alerting finding that nearly 20% of females and 13% of males never consume legumes (Appendix A). 

According to the results of our study, meat represents an important source of protein in the diet of Slovenian adolescents. Males consume more meat than females, and the amount gradually increases with age. The highest usual daily intake of meat was found among males aged 15–17 years (188.5 ± 81.2 g/day). Red meat is more prevalent in males’ diets, while females’ diets show a slightly higher intake of white meat. The study also revealed that males in all age groups consume more meat products than females, with up to twice the amount consumed by their female peers. The results show that a higher percentage of females (7.4%) never consume meat products compared to males (2.1%) (Appendix A). Males exhibit a higher consumption of fish and fish products than females, with only 7.6% reporting no fish consumption, whereas among female adolescents, the percentage of non-consumers of fish was 10.9% (Appendix A).

Considering the widespread consumption of sugar-containing soft beverages (SCSBs), study findings indicate that the highest quantities of these beverages are consumed by males (13–14 years: 128 ± 281 mL/day and 10–12 years: 108 ± 264 mL/day). The data show that 5% of males aged 10–13 years consume over 1220 mL of SCSBs per day (Table 3). The study highlights the popularity of high-sugar foods among Slovenian adolescents, especially evident in males aged 13–14 years, with an average intake of 172 ± 166 g/day (Table 3). 

Table 4 and Table 5 present the proportion of the study population meeting the healthy and sustainable food groups dietary recommendations based on the optimization analysis for the Dutch adolescent population between 9–13 and 14–18 years of age conducted by Brink et al. (2019) [15].

The usual intakes of food groups of Slovenian adolescents notably diverge from Dutch dietary guidelines [15]. Major issues include low intakes of vegetables, milk and dairy products, nuts and seeds, and legumes, while there is an excessive intake of animal-based foods, particularly of red meat (and products) and eggs, surpassing the recommended levels.

In females, the most significant inadequacy in achieving the recommended values is among those aged 14 to 17 years, specifically for vegetables, where 99.2% of individuals do not reach the recommended intake of 250 g/day, followed by milk and dairy products (97.2%) (450 g/day), legumes (92.9%) (120 g/week), and nuts and seeds (90.8%) (25 g/day). In the younger group of females (10–13 years), 98.3% did not reach the recommended intake for milk and dairy products of 450 g/day, 96.6% for legumes (120 g/week), 92.6% for nuts and seeds (25 g/day), and 90.1% for vegetables (150–200 g/day) (Table 4).

Among males aged 14–17 years, 100% of individuals do not reach the recommended lower limit for vegetable intake (250 g/day), 99.1% for milk and dairy products (600 g/day), 94.7% for legumes (120–180 g/week), and 93% for nuts and seeds (25 g/day). In the younger age group of males (10–13 years), 95.9% do not meet the recommendations for milk and dairy products (450 g/day), followed by legumes (95.5%) (120 g/week), nuts and seeds (94.3%) (25 g/day), and vegetables (90.2%) (150–200 g/day) (Table 5).

The LASSO model was applied to assess the influence of certain sociodemographic, anthropometric, and other confounding variables to determine their effects on the change in consumption of 18 food groups in the general adolescent population (detailed in Appendix A). The LASSO model’s tuning parameter, lambda, was optimized using cross-validation to achieve sparsity and identify variables with minimal predictive power for consumption changes. This approach resulted in a variable selection process, retaining only those with nonzero coefficients. Examining the impact on individual food groups, the model identified seven important predictors for the sugar and confectionery group, six for animal foods, five for pasta and rice, and four for bread and bakery products. Sex emerged as the sole predictor for changes in cereal and cereal product consumption. Red meat consumption exhibited the strongest influence from sex, BMI, and season. Overall, season (period, when data were collected) was the most frequent predictor, affecting the consumption of eleven food groups. Sex and cohesion region followed, impacting consumption in nine and eight food groups, respectively. Age group, BMI, and degree of urbanization were also identified as relevant predictors for a subset of food groups.

The data on energy and nutrient intakes and the proportion meeting recommendations are evident from Table 6 and Table 7. Carbohydrates represent 47.0–50.9% of TEI in adolescents, protein 17.0–18.4%, and total fat 24.5–26.5%. The majority of adolescents do not achieve the recommended carbohydrate and fat intake, while also exceeding protein recommendations. Total sugars contribute 15.1–15.7% of the energy intake in males and 16.2–19.2% in females. On the other hand, many adolescents, especially females, consume insufficient dietary fibre (Table 6). Study data revealed that 87% of females (Table 8) and 82% of males aged 15–17 years (Table 9) do not meet the recommendations for dietary fibre intake (at least 21 g/day) [38].

Vitamin D intake is generally low across all age groups among Slovenian adolescents, regardless of sex. Folate intake falls below recommendations, particularly in females aged 13–17 years (Table 6). Vitamin B12 intake is generally adequate but is lacking among females aged 13–17 years. We observed that 50% of females in all age groups fail to meet the lower limit of recommendations (Table 8). Calcium intake is insufficient across all sex and age groups, with over 50% of the population not meeting recommendations; particularly low calcium intake was observed in females (Table 8).

## 4. Discussion

The dietary intakes of Slovenian adolescents do not differ considerably from their peers in some other European countries [40,41,42,43]. In general, adolescents in Slovenia do not meet the Dutch FBDG for the intake of vegetables, milk and dairy products, nuts and seeds, legumes, and water. On the other hand, they exceed the guidelines for the intake of meat and meat products (particularly red meat), high-sugar food, and cheese, and consequently, their dietary pattern is not in line with sustainable dietary recommendations. Compared to D-A-CH reference values, Slovenian adolescents consume insufficient amounts of some nutrients (e.g., vitamin D, calcium, folate among females) that play an essential role in the process of growth and development [15]. However, adolescents generally have sufficient protein intake, primarily from meat, which contradicts the recommendations of a sustainable diet emphasizing a greater consumption of legumes and other plant-based protein sources, such as nuts, seeds, and grains [11,14].

Dietary intakes and eating habits of adolescents pose a significant challenge, as it is necessary to consider various national, specific aspects related to dietary intake both at home and in the school environment. In Slovenia, we have a well-organized school meal system that provides children with 20–50% of their TEI (morning snack and lunch), or even more if the child also consumes other daily meals at school (breakfast and afternoon snack are also available). The school meal system in Slovenia is regulated by law [44], whereas educational institutions are provided with human, financial and spatial resources, which enable the planning, preparation, and distribution of school meals. The guidelines for healthy eating in educational institutions are mandated by law [44]; these institutions are obliged to adhere to them when planning and preparing meals. Recently revised national school dietary guidelines are based on the concept of an optimized mixed diet, which already includes some aspects of sustainability [45]. Guidelines also cover various aspects related to the planning, procurement, and preparation of school meals, including sustainability measures focused on economical food management, meal planning, sourcing locally produced foods, packaging management, food waste reduction, etc. [46]. The results from the national survey on the quality of school meals offered to adolescents aged 10 to 13 indicate that school mid-morning snacks generally adhere to the guidelines, although they tend to have excessive total sugar content. Conversely, more discrepancies were noted in the composition of the school lunches, characterized by insufficient carbohydrates and fats as well as excessive salt content [47,48,49]. The protein content (12.4–13.0 g/snack and 24.3–26.9 g/lunch) and dietary fibre (4.2–4.7 g/snack and 8.2–8.5 g/lunch) in school meals [47,48,49] were in line with the national school dietary guidelines [45]. The school meal system in Slovenia serves as a crucial corrective in the diet of adolescents, particularly for those individuals coming from socioeconomically disadvantaged environments.

As previously mentioned, in children and adolescents, their dietary intakes and habits are notably influenced by their families. Gregorič et al. (2022) point out that typical diets among adults in Slovenia are unbalanced due to a high consumption of meat and meat products, foods high in sugar, fat, and salt, as well as a low intake of fruits, vegetables, milk, and dairy products [19]. Consequently, the proportion of energy from carbohydrates, proteins, and to some extent, free sugars and total fats, as well as the intake of dietary fibre and total water, deviate from the reference values. 

Insufficient vegetable intake is widespread among adolescents, and the issue increases with the age. Nearly 90% of both males and females do not achieve the Dutch recommended daily vegetable intake (150–200 g/day for ages 10–13 years and 250 g/day for ages 14–17 years) [15]. The WHO collaborative cross-national HBSC (Health Behaviour in School-aged Children) survey indicates that among adolescents aged 11–15 years, only 48% consume fruits daily, and just 38% eat vegetables daily. The latest 2022 HBSC study for Slovenia indicates that just 35.8% of adolescents aged 11–17 years consume fruits, and just 35.6% of adolescents consume vegetables daily. Since the previous survey in 2018, there has been a decrease in the percentage of 17-year-olds, both males and females, who regularly consume fruits. Similar findings were also observed with regard to vegetables [50]. This aligns with the WHO data, which states that the frequency of consuming fruits and vegetables decreases with the age of adolescents [51]. Family plays a crucial role in shaping children’s eating habits [52], as the failure to meet vegetable intake recommendations was also identified in the Slovene adult population [19]. The proportion of those meeting the Dutch dietary recommendations for fruits compared to vegetables is more favourable, as approximately 50% to 60% of the Slovene adolescents even exceed the daily intake recommendations. 

It is well known that dietary fibre in whole grains, vegetables, fruits, nuts, and legumes has beneficial health effects [53]. It contributes to reducing the risk of cardiovascular diseases, atherosclerosis and stroke [54,55], hypertension [56], type 2 diabetes, cancer [57,58], and other diseases. Dietary fibre plays an important part in the composition of gastrointestinal microbiota and positively impacts the microbial profile of the digestive tract in children and the production of anti-inflammatory short-chain fatty acids [59]. Results of our study indicate that the average daily intake of dietary fibre among adolescents from all food ranges from 18 to 22 g/day; the majority of Slovenian adolescents have a fibre intake below D-A-CH reference values. As previously highlighted by Koroušić et al. [60], a significant determinant for inadequate dietary fibre among adolescents was sex, and the main foods contributing to dietary fibre intake were bread and other grain products, vegetables, and fruits.

In relation to the low consumption of vegetables, fruits, and grains, we observe that folate intake among Slovenian adolescents falls below recommendations (240 μg/day: 10–12 years; 300 μg/day: 13–17 years) [38]. The proportion of the study population not meeting D-A-CH reference values for folate intake is high, exceeding 50% in males aged 13–17, and over 70% in females of the same age category. 

Milk and dairy products represent an important source of proteins and specific micronutrients, particularly calcium [61,62,63]. In comparative terms, the adolescents in our study on average consumed less milk than their peers in Ireland [42], and similar to our findings, a higher rate of meeting milk intake recommendations in males was also observed in the HELENA study [64]. Findings from cross-sectional studies and longitudinal data suggest that the milk consumption trend decreases with age [65,66]. Similar trends are also indicated by the results of our study. This can be concerning because some authors suggest that reduced milk intake might lead to a higher consumption of soft drinks, which could contribute to childhood obesity [65]. 

Our study also revealed insufficient vitamin D intake, as previously reported [67,68]. We observe a similar trend with calcium intake as well, especially among females, which could be associated with inadequate intake of milk and dairy products in our study, compared to Dutch recommendations. Important dietary sources of calcium include milk and dairy products, selected vegetables like spinach, purslane, chard, endive, and broccoli, legumes, nuts, fish with soft bones (e.g., tinned sardines), and calcium-fortified foods. The majority of calcium in the human body is found in bones and teeth, where it serves a structural role, while the remainder is located in extracellular fluids, intracellular structures, and cell membranes, where it plays a significant role in vascular, neuromuscular, and endocrine functions [69]. Romero-Marco et al. (2020) reported that adolescents (10–17 years old) in Spain, similar to our study, have a lower calcium intake (males: 808.6–1147.0 mg/day; females: 708.0–949.2 mg/day) than recommended [70]. A study conducted by Diethelm et al. (2014), which included adolescents from ten European cities, showed slightly lower calcium intakes (females 10–13 years: 809.0 mg; 13–15 years: 727.0 mg; and 15–19 years: 661 mg; males 10–12 years: 859.0 mg; 13–15 years: 937.0 mg; and 15–19 years: 857.0 mg) [41] compared to our study. Salamoun et al. (2005) reported similar findings, observing both calcium and vitamin D deficiency in adolescents aged 10 to 16 years [71].

To ensure an adequate and sustainable food supply, the EAT-Lancet Commission Summary Report suggests that there will be a need for a shift in people’s dietary patterns in the future, with a greater emphasis on consuming grains, fruits, vegetables, nuts, and legumes in order to achieve not just healthy, but also more sustainable diets [11]. Due to the global trend of environmental consciousness, contemporary guidelines in some countries already place a stronger focus on a plant-based diet [14], which encourages the consumption of plant protein sources, such as legumes, to partially replace proteins from animal origin [72]. The findings of our study indicate that more than 95% of the Slovenian adolescents aged 10 to 17 years do not meet the recommended intake of legumes, which is set at 120 g/week [15]. With a usual daily mean legume intake of approximately 11 g, Slovenian adolescents are far below the EAT-Lancet targets of optimal daily consumption of 75 g of legumes daily. Furthermore, 19.6% of females and 13.4% of males do not consume legumes at all. The determinants expected to be linked with the consumption of legumes are the season, and primarily the place of residence of the adolescents (Appendix A). 

Slovenian adolescents (particularly males) consume excessive amounts of meat (both fresh and processed) (more than 90% of adolescents) as recommended in Dutch guidelines (70 g/day) [15], which could be associated with high protein intakes in our study. A similar situation is also evident with red meat, where the recommended intake limit is even lower (45 g/week), while according to EAT-Lancet Commission Summary Report, only 14 g daily is recommended [11]. Key determinants linked with the consumption of red meat among Slovenian adolescents are sex, age, BMI, and season (Appendix A). 

Despite the fact that red meat represents a major source of iron, a low intake of this nutrient globally remains one of the major public health issues and is usually controversial when it comes to more sustainable diets. It is estimated to affect 47% of preschool and 25% of school-age children around the world [73]. Iron deficiency can result from inadequate intake, increased turnover, or excessive loss (e.g., menstruation) [73]. According to previous studies in Slovenia, the highest prevalence of haemoglobin anaemia is observed among females aged 51–64 years, while critically depleted iron stores have been particularly identified in premenopausal women [74]. Based on the data from our research, the iron intake among adolescents in Slovenia is mostly not problematic, as the majority of the population exceeds the recommended daily intake for iron. The proportion of the population exceeding the recommendation for iron intake is slightly higher in males, as expected. It was found that approximately half of the females reach the recommended value for iron intake (15 mg/day), while the intake slightly decreases with increasing age. In the age group of 15 to 17 years, iron intake is below the recommended value in as many as 66.3% of females. Lavriša et al. (2022) highlighted that the main sources of dietary iron intake are bread and other grain products, and meat and meat products [74]. 

Similar to iron, intakes of vitamin B12, magnesium, and phosphorus in general do not present a dietary concern among Slovenian adolescents.

The data regarding the proportion of the population meeting the weekly recommendation for fish intake (100 g/week) [15] varies slightly depending on the age and place of residence of adolescents (Appendix A). Approximately half of the adolescents do not meet the recommendation, while the other half even exceeds the recommendation for fish intake, which is an encouraging finding. Similar results were observed in adults [19].

Based on sex differences, females were found to have higher sugar intakes compared to males. The latter is also evident from the review of current dietary surveys conducted by Rippin et al. (2019) [75]. As described already by Zupanič et al. (2020), the three main sources of free sugars in Slovenian adolescents’ diets were beverages, bakery products, and dairy products. Among beverages, soft drinks and fruit and vegetable juices were the largest contributors to free sugars intake [76]. Based on the proportion of energy intake from sugar, Slovenian adolescents (15.1–19.2% of TEI) fall among countries reporting lower intakes (Italy: 15.4% of TEI); a notably higher intake of sugar was reported in some countries (Germany: 29.6%; Netherlands: 26.8% of TEI) [77].

Our study findings also indicate that various socioeconomic factors are linked with dietary habits and food consumption. Disparities related to these factors manifest within different food groups. Our data suggest that the season has the most notable impact, followed by sex, cohesion region, age groups, BMI, and place of residency. The data also reveal that the Slovenian adolescents’ dietary intake is not significantly affected by physical activity as assessed by IPAQ and household composition (Appendix A).

In 2023, Slovenia adopted new guidelines for healthy school meals. These guidelines are based not only on the practical needs of the organized school meals system but also on the dietary habits of children and adolescents and the aim to ensure the highest possible level of their health. Our study’s data reveal significant deviations from recommendations in adolescents’ diets, making it crucial for educational institutions to consider these eating habits alongside the dietary guidelines. The data can also serve as an important resource for health education providers, which are systematically administered by health care centres in Slovenia, as well as for teachers in schools.

Some study strengths and limitations should be noted. The strength of the study lies primarily in the use of comprehensive nationally representative data collected by using the EFSA EU Menu project, which follows an international methodology for food consumption surveys. To ensure national representativeness, study subjects were sampled from all regions in Slovenia, using a randomised selection from the Central Population Registry of the Statistical Office of the Republic of Slovenia. Another study strength is the use of the methodology that includes monitoring dietary intake in different seasons using the 24 h dietary recall method, as well as the use of an FPQ. An additional strength is that we compared dietary intake data with recommendations for a sustainable diet, which has become a focus of research due to the impact of eating habits on the environment. However, some limitations should also be mentioned. Slovenia does not have its own food-based dietary guidelines. For this reason, we used recommendations that were developed in other countries. Despite the fact that Slovenia has placed great emphasis on sustainable food production since the adoption of its first national food and nutrition action plan in 2005 [78], we have not yet developed national food-based dietary guidelines that specifically consider global findings in the field of sustainable dietary goals. 

## 5. Conclusions

The study demonstrates that the diet of Slovenian adolescents aged 10 to 17, much like their peers in other developed European countries, diverges from dietary recommendations. Our study highlights some major dietary challenges among Slovenian adolescents. Their food consumption is not in line with sustainable dietary guidelines (e.g., the Netherlands, EAT-Lancet). Specifically, Slovenian adolescents tend to have below-recommended intakes of vegetables, milk and dairy products, nuts, seeds, legumes, and water. Conversely, they often exceed dietary recommendations for meat and meat products, particularly for red meat, as well as sugar and cheese consumption. Moreover, our study has underscored notable disparities in meeting the requirements for certain crucial nutrients (e.g., vitamin D and calcium) that are vital during adolescent growth and development, yet remain unmet by current dietary patterns. Our research indicates that in the process of developing national FBDG for children and adolescents, a broader spectrum of factors should be considered, as various factors influence the selection of different food groups for them (e.g., season, sex, cohesion region, age, BMI, and place of residence). Due to unhealthy dietary habits, it will be necessary in the future to consider a timeframe that enables the achievement of environmental and sustainability goals in dietary guidelines, as well as to ensure the transition of agricultural production into more sustainable forms.

## Figures and Tables

**Table 1 nutrients-16-01912-t001:** Characteristics of adolescent participants in the Slovenian national SI.Menu dietary survey (2017–2018, *n =* 468).

Variables	Levels	All (10–17 Years)	Group 1 (10–12 Years)	Group 2 (13–14 Years)	Group 3 (15–17 Years)
		*n* = 468	*n =* 194	*n =* 93	*n =* 181
Total (%)		495 (100)	201 (100)	100 (100)	194 (100)
Excluded (% of all included in study)		27 (5.5)	7 (3.5)	7 (7.0)	13 (6.7)
Sample for analysis (% of all included in study)		468 (94.5)	194 (96.5)	93 (93.0)	181 (93.3)
Cohesion region of Slovenia—*n* (%) *	Eastern	290 (62.0)	112 (57.7)	66 (71.0)	112 (61.9)
	Western	178 (38.0)	82 (42.3)	27 (29.0)	69 (38.1)
Sex—*n* (%)	Female	238 (50.9)	98 (50.5)	51 (54.8)	89 (49.2)
	Male	230 (49.1)	96 (49.5)	42 (45.2)	92 (50.8)
Degree of urbanization (DEGURBA)—*n* (%)	Rural	122 (26.1)	52 (26.8)	27 (29.0)	43 (23.8)
	Semi-urban	76 (16.2)	34 (17.5)	13 (14.0)	29 (16.0)
	Urban	270 (57.7)	108 (55.7)	53 (57.0)	109 (60.2)
Height (cm)—mean (SD)	Male	166.5 (14.0)	153.8 (9.8)	170.3 (8.2)	178.4 (7.1)
	Female	160.1 (9.1)	153.8 (9.3)	162.6 (5.1)	165.6 (5.7)
Weight (kg)—mean (SD)	Male	60.2 (18.1)	46.9 (11.6)	63.0 (14.3)	73.1 (15.6)
	Female	53.5 (13.1)	46.7 (12.4)	54.3 (10.8)	60.3 (11.0)
** BMI (kg/m^2^)—mean (SD)		21.0 (4.2)	19.6 (3.8)	21.1 (4.1)	22.4 (4.2)
*n* (%)	Underweight (IQR: 5%)	26 (5.6)	10 (5.2)	6 (6.5)	10 (5.5)
	Normal	377 (80.5)	156 (80.4)	71 (76.3)	150 (82.9)
	Overweight or obese	65 (13.9)	28 (14.4)	16 (17.2)	21 (11.6)
Self-assessed IPAQ—*n* (%)	Low intensity	108 (23.3)	44 (22.9)	22 (23.9)	42 (23.5)
	Moderate	141 (30.5)	55 (28.6)	22 (23.9)	64 (35.8)
	High intensity	214 (46.2)	93 (48.4)	48 (52.2)	73 (40.8)
BMR—mean (SD)		1558 (236.2)	1398 (164.5)	1600 (191.0)	1708 (214.0)
EI/BMR—mean (SD)		1.32 (0.38)	1.44 (0.37)	1.32 (0.36)	1.19 (0.37)
Household composition—*n* (%)	Adolescents living with adults	418 (89.3)	179 (92.3)	83 (88.0)	156 (86.2)
	Adolescents living with adults and elderly	50 (10.7)	15 (7.7)	10 (12.0)	25 (13.8)

y—Years; *—Republic of Slovenia is divided into two cohesion regions (Western Slovenia and Eastern Slovenia); SD—standard deviation; cm—centimetres; kg—kilograms; **—BMI reference for underweight [27]; *n*—numerus; IQR—interquartile range; IPAQ—International Physical Activity Questionnaire; BMR—estimated basal metabolic rate [30,31]; EI—estimated intake.

**Table 2 nutrients-16-01912-t002:** Descriptive statistics of usual daily intakes of food groups among female adolescents aged 10–17 years in the SI.Menu study.

Food (Sub)Groups ^#^	10–12 Years						13–14 Years						15–17 Years					
	Mean (SD)	Median	Percentile				Mean (SD)	Median	Percentile				Mean (SD)	Median	Percentile			
			5	25	75	95			5	25	75	95			5	25	75	95
**Milk** (mL/day)	124.8 (106.4)	144.7	0.0	0.0	205.1	298.5	126.4 (102.9)	140.3	0.0	0.0	217.5	267.3	105.9 (149.6)	48.7	0.0	0.0	193.5	280.6
**Dairy products** (yogurt, cheese, milk, cream) (g/day)	64.7 (92.2)	36.8	4.9	25.5	177.7	361.6	75.1 (83.9)	48.6	0.0	27.8	85.6	258.2	53.3 (90.6)	32.0	0.0	0.0	53.3	284.6
Cheese (g/day)	29.8 (27.9)	30.1	0.0	0.0	41.8	96.4	32.9 (23.5)	33.6	0.0	0.0	51.0	65.5	28.6 (24.5)	29.8	0.0	0.0	49.2	69.9
**Vegetables** (fresh and preserved/canned) (g/day)	113.7 (47.5)	106.3	53.8	80.5	141.1	223.4	105.5 (54.1)	100.4	38.2	64.7	142.1	205.2	100.7 (55.7)	91.7	25.0	63.9	130.8	198.4
Fresh vegetables (g/day)	86.4 (35.4)	82.7	35.6	61.3	106.7	156.4	80.9 (48.3)	71.4	32.8	44.7	103.6	154.5	81.8 (49.1)	76.1	0.0	46.5	99.4	186.2
Preserved and canned vegetables (g/day)	23.1 (23.1)	21.7	0.0	0.0	35.7	73.3	22.4 (24.1)	19.8	0.0	0.0	37.2	63.8	17.0 (18.8)	18.5	0.0	0.0	29.4	48.0
**Fruits** (fresh, canned, dry)	200.7 (133.9)	186.1	0.0	124.9	269.6	428.1	132.7 (133.4)	137.6	0.0	0.0	204.5	342.5	166.7 (126.9)	163.1	0.0	87.2	223.4	394.6
Fresh fruits (g/day)	173.2 (119.8)	170.8	0.0	122.6	223.1	369.4	110.1 (103.2)	119.4	0.0	0.0	189.3	307.7	155.0 (119.5)	149.4	0.0	75.8	210.0	346.2
Other fruits (canned, dry) (g/day)	20.8 (57.5)	0.0	0.0	0.0	0.0	167.5	22.6 (58.3)	0.0	0.0	0.0	0.0	131.1	10.4 (31.3)	0.0	0.0	0.0	0.0	87.9
**Legumes** (kidney beans, green beans, lentils, etc.) (g/day)	11.8 (24.7)	0.0	0.0	0.0	0.0	59.5	11.1 (24.7)	0.0	0.0	0.0	0.0	59.9	8.1 (24.4)	0.0	0.0	0.0	0.0	71.0
**Nuts and seeds** (g/day) **	5.5 (24.6)	0.0	0.0	0.0	0.0	32.3	10.7 (22.9)	0.0	0.0	0.0	0.0	60.1	9.6 (29.2)	0.0	0.0	0.0	0.0	56.7
**Potatoes** (g/day)	83.7 (58.1)	89.9	0.0	28.1	127.7	174.0	92.1 (49.9)	106.1	0.0	73.8	128.1	145.9	79.6 (60.9)	99.2	0.0	0.0	127.3	159.5
**Bread and bakery products** (all types of breads, dough, bread-based products) (g/day)	162.1 (84.5)	155.4	72.9	94.2	193.5	312.0	154.6 (64.7)	145.5	76.0	103.4	201.3	263.5	147.4 (104.4)	117.2	0.0	76.8	207.4	348.2
Bread (g/day)	121.4 (59.5)	113.2	50.3	79.6	152.9	251.8	135.3 (61.5)	134.9	35.9	99.3	170.9	229.9	101.7 (67.5)	88.9	0.0	64.7	136.4	206.1
**Cereal and cereal products** (breakfast cereals, dry pasta, rice, etc.) (g/day)	120.6 (69.6)	116.1	14.5	70.1	166.7	255.0	120.6 (95)	88.9	12.9	49.0	171.4	292.1	109.1 (99.8)	85.5	0.0	38.3	161.4	298.5
Breakfast cereals (g/day)	58.5 (58.9)	39.4	0.0	17.1	82.6	167.0	60.5 (69.5)	35.0	0.0	18.5	75.8	215.3	55.8 (80.7)	28.8	0.0	0.0	69.8	194.0
Dry pasta, dry rice (g/day)	62.1 (46.8)	57.7	0.0	28.7	91.8	166.7	60.2 (53.7)	50.2	0.0	0.0	94.7	147.6	53.3 (51.6)	49.2	0.0	0.0	91.8	143.8
**Fish and fish products** (fresh fish, canned fish, etc.) (g/day)	15.2 (31.1)	0.0	0.0	0.0	0.0	87.5	21.9 (49.5)	0.0	0.0	0.0	0.0	133.3	21.0 (54.7)	0.0	0.0	0.0	0.0	132.0
Fresh fish (g/day)	5.6 (24.5)	0.0	0.0	0.0	0.0	71.8	16 (46.7)	0.0	0.0	0.0	0.0	133.3	12.3 (50.8)	0.0	0.0	0.0	0.0	70.9
**Fresh meat** (red meat and poultry) (g/day)	133.0 (58.7)	132.6	54.8	85.7	167.1	204.9	123.2 (58.8)	136.2	44.7	71.5	172.4	201.0	134.0 (67.4)	136.5	41.1	83.0	168.7	234.2
Red meat (g/day)	70 (39.5)	63.4	0.0	50.0	79.7	143.5	62 (36)	61.4	0.0	46.9	77.9	138.5	63.1 (43.8)	61.5	0.0	39.1	86.1	145.3
Poultry (g/day)	63.7 (49)	73.4	0.0	0.0	102.1	133.7	64.2 (45.6)	76.7	0.0	0.0	97.4	123.3	69.7 (51.6)	85.3	0.0	0.0	108.2	139.9
**Processed meat** (sausages, salami, and other processed meat) (g/day)	29.3 (44.1)	0.0	0.0	0.0	76.1	113.7	37 (44.6)	0.0	0.0	0.0	79.7	110.9	21.0 (38.3)	0.0	0.0	0.0	28.0	110.9
**Fruit and vegetable juices** (mL/day)	90.5 (132.6)	0.0	0.0	0.0	194.3	341.3	118.4 (146)	45.9	0.0	0.0	221.4	298.1	129.6 (185.4)	0.0	0.0	0.0	221.4	503.4
**Sugar-containing soft beverages** (mL/day)	69.9 (171.7)	0.0	0.0	0.0	0.0	527.0	84.1 (195.4)	0.0	0.0	0.0	0.0	499.4	86.7 (199)	0.0	0.0	0.0	0.0	475.0
**Tap and bottled water** (mL/day)	696.1 (382.1)	655.9	129.2	444.4	900.1	1439.6	694 (292.4)	664.8	252.4	466.0	905.7	1228.6	749.6 (439.3)	699.9	176.7	370.6	1024.2	1665.1
**Hot drinks** (coffee, tea, cacao and hot chocolate, etc.) (mL/day)	265.4 (254.9)	227.6	0.0	108.1	368.6	753.3	292.6 (267)	242.1	0.0	113.9	366.5	629.5	282.5 (226.8)	248.4	0.0	108.5	410.0	692.7
Tea (mL/day)	139.5 (221.9)	0.0	0.0	0.0	247.2	528.7	169.4 (249.4)	57.2	0.0	0.0	251.2	579.6	160 (207.1)	60.4	0.0	0.0	254.1	485.9
Coffee drinks (mL/day)	126.2 (79.7)	108.5	0.0	84.8	153.1	286.9	109 (72.7)	110.5	0.0	59.8	160.3	237.1	110.5 (94.3)	107.7	0.0	0.0	155.0	285.5
**Fats and oils** (vegetable oils, margarines, butter and other animal fat) (g/day)	17.7 (11.6)	14.9	4.3	10.7	22.2	42.2	19.1 (11.8)	15.7	5.8	10.8	23.0	44.5	17.6 (10)	14.4	6.3	10.9	22.9	37.6
Vegetable oils and fats (g/day)	12.8 (6)	12.5	6.0	8.7	15.6	24.5	13.2 (5.2)	13.1	6.5	9.0	17.0	22.1	14.4 (6.3)	13.0	6.9	10.2	17.4	26.5
Butter and other animal fat (g/day)	4.5 (7.8)	0.0	0.0	0.0	6.9	23.0	6.4 (10.2)	0.0	0.0	0.0	11.8	27.0	3.3 (7)	0.0	0.0	0.0	0.0	19.7
**High sugar food** (sugar, confectionary, cakes, cookies, desserts) (g/day)	129.8 (128.8)	96.8	0.0	26.9	202.0	396.1	119.3 (116.7)	87.5	4.5	31.7	177.3	302.6	124.0 (124.4)	85.1	0.0	30.4	192.0	360.9
Sugar and confectionary (g/day)	30.9 (34.1)	19.6	0.0	0.0	51.3	97.9	34 (31.1)	28.9	0.0	10.7	49.5	95.4	30.5 (30.2)	22.3	0.0	0.0	46.3	86.6
Cakes, cookies (g/day)	27.8 (46.3)	0.0	0.0	0.0	40.8	114.2	28.4 (49.8)	0.0	0.0	0.0	60.7	157.7	37.4 (61.8)	0.0	0.0	0.0	57.9	162.2
Desserts (g/day)	71.1 (107.3)	0.0	0.0	0.0	106.9	318.4	56.9 (100.7)	0.0	0.0	0.0	77.1	290.0	56.0 (97.5)	0.0	0.0	0.0	101.0	245.3
**Fresh and food-incorporated eggs** * (g/day)	32.6 (35.4)	31.5	0.0	0.0	53.4	101.4	28.4 (36.6)	0.0	0.0	0.0	54.1	90.8	35.8 (35.8)	33.2	0.0	0.0	62.9	99.4
**Ready-to-eat meals** (g/day)	45.2 (56.1)	29.1	0.0	0.0	72.0	164.2	41.6 (51)	29.1	0.0	0.0	72.2	139.3	27.7 (39.5)	0.0	0.0	0.0	52.1	100.8

Note: * Habitual intake of eggs calculated from 2 × 24 HDRs. ** Data for nuts and seeds. ^#^ Subcategory “Other” not presented.

**Table 3 nutrients-16-01912-t003:** Descriptive statistics of usual daily intakes of food groups among male adolescents aged 10–17 years in the SI.Menu study.

Food (Sub)Groups ^#^	10–12 Years						13–14 Years						15–17 Years					
	Mean (SD)	Median	Percentile				Mean (SD)	Median	Percentile				Mean (SD)	Median	Percentile			
			5	25	75	95			5	25	75	95			5	25	75	95
**Milk** (mL/day)	157.3 (118.6)	167.7	0.0	44.1	232.8	359.6	166.0 (126.4)	199.2	0.0	35.8	247.0	389.1	163.7 (144.8)	167.7	0.0	38.0	253.1	461.6
**Dairy products** (yogurt, cheese, milk, cream) (g/day)	75.2 (90.4)	37.9	0.0	19.0	99.7	285.3	38.6 (36.8)	33.8	0.0	0.0	54.4	75.1	73.4 (90.2)	47.2	0.0	20.3	85.1	308.1
Cheese (g/day)	31.3 (25.4)	32.2	0.0	0.0	45.8	78.2	29.3 (25)	33.3	0.0	0.0	45.7	90.4	38.2 (29.3)	41.8	0.0	17.3	54.8	77.0
**Vegetables** (fresh and preserved/canned) (g/day)	105.6 (48.8)	98.8	34.2	71.9	139.5	207.9	110.6 (48.4)	102.8	36.1	78.8	130.4	275.3	114.7 (58.2)	99.6	42.7	76.1	141.3	229.8
Fresh vegetables (g/day)	78.2 (40.9)	73.8	32.3	46.2	100.7	155.5	81.0 (43.5)	72.6	34.4	52.1	104.1	256.3	86.2 (48.1)	78.4	31.7	53.8	110.7	167.6
Preserved and canned vegetables (g/day)	25.5 (26)	23.2	0.0	0.0	34.6	79.7	29.5 (26.7)	26.4	0.0	0.0	39.0	114.5	26.2 (27.6)	25.4	0.0	0.0	34.9	67.6
**Fruits** (fresh, canned, dry)	184.6 (120.3)	194.6	0.0	105.7	279.5	372.7	160.2 (139.4)	152.4	0.0	0.0	209.4	606.9	117.7 (131.7)	92.5	0.0	0.0	204.0	327.0
Fresh fruits (g/day)	165.5 (110.2)	176.9	0.0	92.5	250.0	355.0	137.5 (122)	146.2	0.0	0.0	208.5	465.3	108.1 (114.3)	92.5	0.0	0.0	190.5	310.0
Other fruits (canned, dry) (g/day)	18.5 (50.5)	0.0	0.0	0.0	0.0	172.4	13.8 (44)	0.0	0.0	0.0	0.0	194.8	10.1 (39.1)	0.0	0.0	0.0	0.0	114.1
**Legumes** (kidney beans, green beans, lentils, etc.) (g/day)	8.0 (21.2)	0.0	0.0	0.0	0.0	53.9	12.4 (23.8)	0.0	0.0	0.0	28.7	105.0	10.6 (27.7)	0.0	0.0	0.0	0.0	78.6
**Nuts and seeds** (g/day) **	6.6 (17.8)	0.0	0.0	0.0	0.0	46.5	2.4 (9.6)	0.0	0.0	0.0	0.0	48.3	9.8 (24.2)	0.0	0.0	0.0	0.0	72.3
**Potatoes** (g/day)	92.7 (67.9)	116.5	0.0	0.0	145.6	182.7	87.8 (65.2)	103.1	0.0	0.0	141.9	183.3	102.6 (73.7)	120.8	0.0	0.0	163.6	201.4
**Bread and bakery products** (all types of breads, dough, bread-based products) (g/day)	205.5 (98.6)	192.6	74.2	134.1	246.4	411.4	216.6 (99.6)	210.9	78.4	139.0	294.1	456.3	228.4 (104.9)	222.2	97.2	149.9	281.2	428.0
Bread (g/day)	160.3 (70.1)	154.0	55.2	101.3	205.8	273.3	168.8 (71)	150.2	74.9	129.2	219.2	313.4	162.7 (74.4)	139.2	69.3	109.0	209.5	299.8
**Cereal and cereal products** (breakfast cereals, dry pasta, dry rice, etc.) (g/day)	139.9 (95.7)	120.5	22.4	75.8	186.6	329.1	162.6 (125.1)	135.2	12.0	69.5	240.4	600.5	142.1 (122.6)	120.7	0.0	48.2	192.2	365.8
Breakfast cereals (g/day)	69.3 (78.1)	39.1	0.0	18.5	89.5	242.6	84.3 (106.4)	52.3	0.0	19.9	121.6	558.4	67.3 (94)	28.5	0.0	9.5	87.1	293.8
Dry pasta, dry rice (g/day)	70.7 (55)	60.8	0.0	24.3	112.7	161.5	78.3 (70.9)	57.1	0.0	23.4	135.2	237.4	74.9 (73.8)	64.1	0.0	0.0	126.6	199.7
**Fish and fish products** (fresh fish, canned fish, etc.) (g/day)	19.7 (44.1)	0.0	0.0	0.0	0.0	134.2	33.7 (54.5)	0.0	0.0	0.0	78.9	207.7	18.3 (54.6)	0.0	0.0	0.0	0.0	107.7
Fresh fish (g/day)	14.0 (38.2)	0.0	0.0	0.0	0.0	127.8	15.1 (45.4)	0.0	0.0	0.0	0.0	207.7	8.6 (42.5)	0.0	0.0	0.0	0.0	0.0
**Fresh meat** (red meat and poultry) (g/day)	152 (56.2)	153.5	66.6	110.3	191.6	245.5	170.9 (69)	170.1	71.4	106.8	219.5	274.0	188.5 (81.2)	191.0	74.5	129.3	243.5	313.7
Red meat (g/day)	83.7 (38.9)	77.8	0.0	65.8	98.5	143.6	97.3 (38.9)	92.8	54.4	73.7	111.5	267.3	106.7 (54.6)	98.4	0.0	81.7	141.8	211.1
Poultry (g/day)	68.3 (55.1)	75.1	0.0	0.0	110.9	156.9	73.6 (66.1)	75.0	0.0	0.0	141.3	191.8	80.8 (64.4)	90.3	0.0	0.0	122.7	176.6
**Processed meat** (sausages, salami, and other processed meat) (g/day)	46.5 (53.2)	0.0	0.0	0.0	100.0	136.1	48.5 (54.2)	0.0	0.0	0.0	105.0	155.0	51.8 (62.4)	0.0	0.0	0.0	113.9	149.7
**Fruit and vegetable juices** (mL/day)	100.7 (147)	0.0	0.0	0.0	228.7	449.6	130.5 (175.4)	55.7	0.0	0.0	228.7	763.4	140.4 (202.7)	0.0	0.0	0.0	230.5	517.9
**Sugar-containing soft beverages** (mL/day)	108.0 (263.5)	0.0	0.0	0.0	76.2	610.6	128.2 (280.5)	0.0	0.0	0.0	0.0	1220.2	78.5 (178.9)	0.0	0.0	0.0	0.0	449.9
**Tap and bottled water** (mL/day)	747.2 (361.8)	705.6	253.0	468.5	979.7	1304.8	938.0 (448.4)	908.3	292.6	558.8	1236.2	2232.5	994.7 (613.8)	883.5	203.1	542.7	1367.5	2212.4
**Hot drinks** (coffee, tea, cacao and hot chocolate, etc.) (mL/day)	320.5 (271.2)	289.8	0.0	109.0	459.4	871.5	256.9 (226.3)	207.3	0.0	107.4	364.4	878.2	277.4 (232.9)	247.2	0.0	129.5	355.2	791.3
Tea (mL/day)	183.8 (222.7)	132.2	0.0	0.0	293.7	715.9	147.1 (191.7)	0.0	0.0	0.0	251.2	487.0	163.8 (199.1)	129.1	0.0	0.0	253.6	555.9
Coffee drinks (mL/day)	168 (133.1)	130.5	29.4	107.4	199.3	370.4	143.5 (62.5)	135.0	50.4	108.1	198.4	241.7	77.3 (79.6)	65.2	0.0	0.0	129.5	226.5
**Fats and oils** (vegetable oils, margarines, butter and other animal fat) (g/day)	20.5 (13.9)	16.8	5.1	9.8	27.9	51.8	23.4 (16.6)	19.7	7.0	13.6	25.9	90.7	23.7 (14.4)	20.4	5.2	13.5	32.0	52.4
Vegetable oils and fats (g/day)	14.4 (7.5)	12.8	6.5	9.8	16.9	29.4	14.9 (5.5)	13.3	7.5	11.8	16.8	36.1	18.0 (8.2)	16.3	7.4	13.0	21.6	33.5
Butter and other animal fat (g/day)	6.9 (10)	0.0	0.0	0.0	12.0	32.7	7.2 (15.3)	0.0	0.0	0.0	10.4	74.1	6.2 (11.3)	0.0	0.0	0.0	10.4	34.4
**High sugar food** (sugar, confectionary, cakes, cookies, desserts) (g/day)	131.4 (120.9)	106.0	0.0	28.4	208.7	393.3	172 (166.1)	146.7	0.0	26.4	284.2	650.3	129.1 (144)	72.3	0.0	7.2	222.3	407.6
Sugar and confectionary (g/day)	35.1 (39.7)	22.8	0.0	0.0	55.3	96.3	42.8 (41.9)	32.8	0.0	0.0	71.5	154.2	30.4 (37.9)	16.9	0.0	0.0	51.5	101.9
Cakes, cookies (g/day)	32.5 (51.4)	0.0	0.0	0.0	54.9	159.1	47.2 (75.8)	0.0	0.0	0.0	75.6	319.0	34.3 (62.7)	0.0	0.0	0.0	50.2	188.1
Desserts (g/day)	63.8 (108.3)	0.0	0.0	0.0	133.3	293.7	82.0 (120.2)	0.0	0.0	0.0	155.9	565.5	64.4 (111.9)	0.0	0.0	0.0	118.3	293.7
**Fresh and food-incorporated eggs** * (g/day)	37.4 (38.8)	33.1	0.0	0.0	66.1	103.8	37.7 (37.1)	34.3	0.0	0.0	63.3	145.8	36.3 (40.1)	32.8	0.0	0.0	67.9	114.6
**Ready-to-eat meals** (g/day)	50.1 (64.6)	30.0	0.0	0.0	71.5	180.4	34.3 (44.8)	9.9	0.0	0.0	57.3	162.1	46.1 (63.4)	19.4	0.0	0.0	71.5	191.6

Note: * Habitual intake of eggs calculated from 2 × 24 HDRs. ** Data for nuts and seeds. ^#^ Subcategory “Other” not presented.

**Table 4 nutrients-16-01912-t004:** Proportion (%) of the female SI.Menu study population meeting recommended values for daily intake of specific food groups.

Food Groups	Females								Recommended Daily Amounts	
	10–13 Years				14–17 Years					
	% Meet DRI	% Do Not Meet DRI	% Below DRI	% Above DRI	% Meet DRI	% Do Not Meet DRI	% Below DRI	% Above DRI	10–13 Years	14–17 Years
Vegetables (fresh and preserved/canned)	6.6	93.4	90.1	3.3	N/A	N/A	99.2	0.8	150–200 g/day	250 g/day
Fruits	N/A	N/A	43.0	57.0	N/A	N/A	38.8	61.2	200 g/day	200 g/day
Bread	31.6	68.4	26.3	42.1	23.8	76.2	66.7	9.5	140–175 g/day	140–175 g/day
Cereal products and potatoes	22.3	77.7	65.3	12.4	6.4	93.6	78.9	14.7	300–500 kcal/day	400–500 kcal/day
Nuts and seeds	N/A	N/A	92.6	7.4	N/AN/A	N/A	90.8	9.2	25 g/day	
Milk and dairy products ^1^	N/A	N/A	98.3	1.7	N/A	N/A	97.2	2.8	450 g/day	450 g/day
Cheese ^2^	N/A	N/A	2.4	97.6	N/A	N/A	48.0	52.0	20 g/day	40 g/day
Water and non-alcoholic beverages ^3^	N/A	N/A	57.4	42.6	N/A	N/A	60.6	39.4	900 mL/day	1000 mL/day
Meat (fresh and processed)	N/A	N/A	5.9	94.1	N/A	N/A	7.0	93.0	70 g/day	
Red meat	N/A	N/A	9.6	90.4	N/A	N/A	8.5	91.5	45 g/day	
Weekly recommended										
Fish and fish products	N/A	N/A	47.1	52.9	N/A	N/A	55.2	44.8	100 g/week	
Legumes	N/A	N/A	96.6	3.4	7.1	92.9	92.9	0.0	120 g/week	120–180 g/week
Eggs ^4^	2.7	97.3	97.3	0.0	4.9	95.1	95.1	0.0	100–150 g/week	

The cut-off values for food groups intakes are adopted from Dutch Healthy and Sustainable FBDG. Recommended daily amounts of food groups provide about 85% of the energy requirement [15]. For ranges, the proportion of the study population with excess/insufficient consumptions is reported. ^1^ Sum of recommendations for “milk and dairy products” and “cheese”. ^2^ Cheeses include all hard cheeses and cheese spreads. ^3^ Water and non-alcoholic beverages refer to bottled and tap water, coffee, tea, and all types of non-alcoholic beverages. ^4^ Eggs recommendation calculated based on data for habitual intake of eggs. Weekly recommendations were converted to daily value; g—grams; mL—millilitres; kcal—kilocalories; N/A—not applicable.

**Table 5 nutrients-16-01912-t005:** Proportion (%) of the male SI.Menu study population meeting recommended values for daily intake of specific food groups.

Food Groups	Males								Recommended Daily Amounts	
	10–13 Years				14–17 Years					
	% Meet DRI	% Do Not Meet DRI	% Below DRI	% Above DRI	% Meet DRI	% Do Not Meet DRI	% Below DRI	% Above DRI	10–13 Years	14–17 Years
Vegetables (fresh and preserved/canned)	8.9	91.1	90.2	0.8	N/A	N/A	100.0	0.0	150–200 g/day	250 g/day
Fruits	N/A	N/A	47.0	53.0	N/A	N/A	38.6	61.4	200 g/day	200 g/day
Bread	38.8	61.2	10.2	51.0	48.0	52.0	20.0	32.0	175–210 g/day	210–280 g/day
Cereal products and potatoes	8.1	91.9	70.7	21.1	3.5	96.5	78.3	18.3	400–500 kcal/day	500–600 kcal/day
Nuts and seeds	N/A	N/A	94.3	5.7	N/A	N/A	93.0	7.0	25 g/day	
Milk and dairy products ^1^	N/A	N/A	95.9	4.1	N/A	N/A	99.1	0.8	450 g/day	600 g/day
Cheese ^2^	N/A	N/A	3.4	96.6	N/A	N/A	31.3	68.7	20 g/day	40 g/day
Water and non-alcoholic beverages ^3^	N/A	N/A	61.0	39.0	N/A	N/A	59.5	40.5	1000 mL/day	1300 mL/day
Meat (fresh and processed)	N/A	N/A	0.8	99.2	N/A	N/A	0.0	100.0	70 g/day	
Red meat	N/A	N/A	1.6	98.4	N/A	N/A	0.0	100.0	45 g/day	
Weekly recommended										
Fish and fish products	N/A	N/A	51.6	48.4	N/A	N/A	31.1	60.9	100 g/week	
Legumes	N/A	N/A	95.5	4.5	5.3	94.7	94.7	0.0	120 g/week	120–180 g/week
Eggs ^4^	11.1	88.9	88.9	0.0	23.4	76.6	72.7	3.9	100–150 g/week	

The cut-off values for food groups intakes are adopted from Dutch Healthy and Sustainable FBDG. Recommended daily amounts of food groups provide about 85% of the energy requirement [15]. For ranges, the proportion of the study population with excess/insufficient consumptions is reported. ^1^ Sum of recommendations for “milk and dairy products” and “cheese”. ^2^ Cheeses include all hard cheeses and cheese spreads. ^3^ Water and non-alcoholic beverages refer to bottled and tap water, coffee, tea, and all types of non-alcoholic beverages. ^4^ Eggs recommendation calculated based on data for habitual intake of eggs. Weekly recommendations were converted to daily value; g—grams; mL—millilitres; kcal—kilocalories; N/A—not applicable.

**Table 6 nutrients-16-01912-t006:** Descriptive statistics of usual daily intakes of energy, macronutrients, dietary fibre, water, and selected micronutrients among female adolescents aged 10–17 years in the SI.Menu study.

	10–12 Years				13–14 Years				15–17 Years			
	Mean (SD)	Median	Percentile		Mean (SD)	Median	Percentile		Mean (SD)	Median	Percentile	
			25%	75%			25%	75%			25%	75%
**Energy kJ/day**	7795 (1693)	7898	6740	8644	7683 (1549)	7357	6449	8984	7212 (2171)	7226	5708	8253
kJ/kg body weight	178.4 (58.5)	174.9	134.2	210.3	148 (47.9)	142.7	117.9	162.4	123.4 (43.8)	116.5	92.0	150.2
**Carbohydrates g/day**	236.8 (56.8)	240.6	207.6	270.6	218.4 (43.9)	216.0	192.9	247.7	214.9 (68.8)	212.4	167.5	253.9
kJ/day	3965 (950)	4030	3477	4532	3658 (735)	3618	3231	4149	3599 (1153)	3558	2806	4253
% TEI	50.9 (6.6)	50.9	46.0	55.1	47.9 (5.7)	47.7	44.1	52.3	50.3 (8.1)	50.0	44.5	57.3
**Total sugars ^a^ g/day**	82 (31.9)	77.8	62.5	99.7	70.6 (26.3)	68.3	50.8	90.6	77 (31.7)	70.5	56.6	98.2
kJ/day	1435 (559)	1362	1093	1745	1236 (460)	1195	889	1586	1349 (555)	1234	991	1720
% TEI	18.3 (5.5)	18.1	14.2	22.0	16.2 (5.5)	15.7	13.1	19.4	19.2 (6.9)	19.8	14.1	22.7
**Free sugars g/day**	49.6 (26.1)	47.7	32.6	60.1	44 (23.4)	38.0	24.6	57.5	46.5 (28.8)	41.2	26.2	65.9
kJ/day	868 (458)	836.0	571.0	1051	770 (410)	665.0	430.0	1007	814 (504)	722	459.0	1154
% TEI	10.9 (5.2)	10.4	7.7	14.1	9.9 (4.4)	9.9	5.5	12.8	11.3 (6.3)	10.8	6.9	14.4
**Proteins g/day**	78 (19.6)	75.0	64.1	91.3	80.7 (19.7)	75.2	69.8	94.5	73.6 (24.3)	75.3	58.1	86.7
kJ/day	1307 (328)	1255	1074	1529	1352 (330)	1260	1169	1583	1232 (406)	1261	974	1453
% TEI	17 (3.5)	16.6	15.2	18.2	17.5 (2.1)	17.2	15.9	19.1	17.2 (3.5)	16.9	15.3	19.4
g/kg body weight	1.8 (0.7)	1.7	1.3	2.2	1.6 (0.6)	1.5	1.2	1.8	1.3 (0.5)	1.2	0.9	1.6
**Total fats g/day**	65.6 (19)	64.1	52.2	77.4	69.9 (19.4)	67.7	59.8	79.5	62.9 (26)	60.5	44.0	77.4
kJ/day	1922 (556)	1880	1531	2269	2048 (568)	1983	1751	2331	1842 (763)	1772	1290	2268
% TEI	24.5 (4.3)	24.1	21.6	27.8	26.5 (4.3)	27.0	23.3	29.3	25.1 (5.5)	24.7	21.3	28.3
**Dietary fibre g/day**	19.9 (5.9)	18.9	15.7	22.8	17.9 (5.6)	16.4	14.8	20.4	18 (8.2)	15.3	12.5	21.6
g/MJ	2.6 (0.8)	2.5	2.1	2.9	2.4 (0.7)	2.3	1.8	2.8	2.5 (0.9)	2.3	1.9	2.9
**Total water ^b^ mL/day**	1843 (500)	1777	1497	2124	1802 (390)	1784	1586	2139	1894 (652)	1772	1421	2235
**Vitamin D (µg/day)**	2.1 (1.1)	1.9	1.5	2.4	2.5 (1.6)	2.0	1.6	3.0	2.8 (2.7)	2.0	1.6	2.9
**Folate (µg/day)**	284.9 (81.2)	275.7	218.0	340.3	269.2 (86.3)	250.5	213.9	303.9	263.7 (110.1)	228.9	186.0	322.6
**Vitamin B12 (µg/day)**	4.7 (2.6)	4.2	3.0	5.6	5.7 (6.8)	4.1	3.3	5.8	4.6 (3.1)	3.8	2.7	5.4
**Fe (mg/day)**	15.1 (4)	14.9	11.8	17.2	14.5 (3)	14.4	12.1	16.5	14.2 (5.3)	13.0	10.6	17.6
**Ca (mg/day)**	874 (300)	833	652.8	1023	919 (303)	869	739.5	1100	894 (378.3)	820	631.4	1052
**Mg (mg/day)**	333.3 (99.4)	327.1	254.4	378.1	332.4 (100)	326.2	270.7	376.4	316.7 (114.6)	274.1	231.8	398.2
**P (g/day)**	1.4 (0.3)	1.3	1.1	1.5	1.4 (0.3)	1.3	1.2	1.6	1.3 (0.4)	1.2	1.0	1.6

Note: ^a^ Total sugars: free sugars and sugars naturally occurring in foods (e.g., lactose in milk, fructose in fruits) [39]. ^b^ Total water refers to the total water intake from beverages and solid foods.

**Table 7 nutrients-16-01912-t007:** Descriptive statistics of usual daily intakes of energy, macronutrients, dietary fibre, water, and selected micronutrients among male adolescents aged 10–17 years in the SI.Menu study.

	10–12 Years				13–14 Years				15–17 Years			
	Mean (SD)	Median	Percentile		Mean (SD)	Median	Percentile		Mean (SD)	Median	Percentile	
			25%	75%			25%	75%			25%	75%
**Energy (kJ/day)**	8757 (1780)	8826	7507	10,064	9639 (2229)	9300	8088	11,100	9687 (2452)	9403	7958	11,658
kJ/kg body weight	198.9 (66.9)	191.4	154.7	245.5	161.8 (55.4)	157.6	119.3	197.9	138.7 (46.5)	138.6	105.7	172.1
**Carbohydrates g/day**	250.9 (56.7)	246.5	204.9	300.6	275.1 (71.1)	274.2	222.4	312.4	270.7 (73.5)	261.1	218.4	333.0
kJ/day	4202 (949)	4129	3431	5034	4607 (1190)	4592	3725	5232	4534 (1231)	4373	3658	5578
% TEI	48.2 (6.7)	49.2	43.7	52.1	47.8 (5.9)	47.9	44.4	52.4	47 (6.7)	46.5	42.5	51.1
**Total sugars ^a^ (g/day)**	78.4 (27.7)	79.0	56.4	100.8	86 (40.5)	81.5	60.4	102.1	85.5 (41.9)	84.5	52.6	114.8
kJ/day	1373 (486)	1383	987	1764	1506 (709)	1426	1057	1788	1497 (733)	1480	921	2010
% TEI	15.7 (4.9)	15.7	13.1	18.7	15.3 (5.1)	15.3	12.2	18.3	15.1 (6.1)	15.3	11.0	19.5
**Free sugars g/day**	48.3 (23.2)	47.2	30.7	62.3	57.5 (38.9)	51.0	29.6	71.7	60 (36.1)	50.0	34.1	88.6
kJ/day	845 (407)	826.0	537	1091	1007 (682)	893	518	1255	1050 (632)	875	598	1550
% TEI	9.6 (4.2)	9.6	6.2	12.3	10 (5.3)	10.2	6.5	13.2	10.5 (5.8)	9.3	6.6	14.8
**Proteins g/day**	91.2 (21.8)	89.7	76.5	105.8	101.7 (24.3)	102.3	88.5	116.4	105.2 (26.8)	102.9	87.9	124.3
kJ/day	1527 (365)	1502	1282	1771	1703 (407)	1713	1482	1950	1761 (449)	1723	1472	2081
% TEI	17.5 (2.6)	17.2	15.5	19.0	17.9 (3.3)	17.1	15.7	20.7	18.4 (3.1)	18.3	16.0	20.5
g/kg body weight	2.1 (0.7)	2.0	1.5	2.5	1.7 (0.5)	1.6	1.3	1.9	1.5 (0.5)	1.5	1.1	1.7
**Total fats g/day**	79.1 (24.4)	78.2	62.2	96.4	86.3 (25.9)	79.8	68.2	105.2	88.4 (30)	85.6	72.3	108.1
kJ/day	2317 (715)	2292	1822	2824	2529 (760)	2340	2000	3084	2590 (880)	2508	2118	3167
% TEI	26.2 (4.8)	26.0	23.1	28.7	26.1 (4)	26.4	22.8	29.1	26.4 (4.8)	26.3	23.2	29.6
**Dietary fibre g/day**	20.7 (7.4)	20.8	15.3	24.7	21.6 (7.7)	20.6	15.5	25.6	19.6 (7.7)	18.8	14.5	23.5
g/MJ	2.4 (0.7)	2.3	1.9	2.7	2.3 (0.7)	2.1	1.8	2.8	2 (0.7)	1.9	1.5	2.4
**Total water ^b^ mL/day**	1996 (470)	1960	1673	2275	2201 (540)	2198	1835	2479	2376 (709)	2327	1821	2889
**Vitamin D (µg/day)**	2.6 (1.4)	2.2	1.7	3.0	3.4 (1.6)	2.9	2.4	4.0	3.3 (1.4)	3.0	2.3	3.8
**Folate (µg/day)**	290.1 (100.5)	278.1	221.8	351.6	326.4 (131.5)	290.8	232.0	394.7	297.1 (133.6)	276.5	211.2	360.9
**Vitamin B12 (µg/day)**	5.4 (5.5)	4.7	3.6	5.8	6.8 (5.6)	5.4	4.2	7.2	6.1 (3.2)	5.3	4.1	7.4
**Fe (mg/day)**	17.2 (5)	16.9	13.7	19.3	20.3 (6.4)	19.4	15.8	22.5	18.1 (5.6)	17.9	13.9	21.6
**Ca (mg/day)**	1027 (315)	1046	787.7	1208	1096 (517)	1057	878.0	1217	1077 (473.6)	1004	759.1	1333
**Mg (mg/day)**	357.1 (113.7)	338.1	264.2	417.8	409.3 (132.2)	383.9	335.4	457.6	375.9 (129.2)	361.7	291.6	444.6
**P (g/day)**	1.50 (0.40)	1.50	1.30	1.80	1.70 (0.40)	1.60	1.50	2.00	1.70 (0.50)	1.60	1.30	2.00

Note: ^a^ Total sugars: free sugars and sugars naturally occurring in foods (e.g., lactose in milk, fructose in fruits) [39]. ^b^ Total water refers to the total water intake from beverages and solid foods.

**Table 8 nutrients-16-01912-t008:** The proportion (%) of the female SI.Menu study population meeting daily dietary reference values of macronutrients and micronutrients.

Macronutrients	Female												Dietary Reference Values (DRVs)		
	10–12 Years				13–14 Years				15–17 Years						
	% Meet DRI	% Do Not Meet DRI	% Below DRI	% Above DRI	% Meet DRI	% Do Not Meet DRI	% Below DRI	% Above DRI	% Meet DRI	% Do Not Meet DRI	% Below DRI	% Above DRI	10–12 Years	13–14 Years	15–17 Years
Carbohydrates	59.4	40.6			31.0	69.0			50.0	50.0			>50% of total energy intake		
Free sugars (criteria 1) ^a^	45.8	54.2			52.4	47.6			42.4	57.6			<10% of total energy intake		
Free sugars (criteria 2) ^a^	12.5	87.5			16.7	83.3			13.0	87.0			<5% of total energy intake		
Proteins (criteria 1)			3.1	96.9			2.4	97.6			13.0	87.0	42 g/day	49 g/day	48 g/day
Proteins (criteria 2)			4.2	95.8			4.8	95.2			13.0	87.0	0.9 g/kg of b. w.		0.8 g/kg of b. w.
Total fats	12.5	87.5	87.5	0.0	16.7	83.3	83.3	0.0			82.6	17.4	30–35% of TEI		30% of TEI
Dietary fibre			71.9	28.1			78.6	21.4			87.0	13.0	at least 19 g/day		at least 21 g/day
Total water ^b^			57.3	42.7			64.3	35.7			67.4	32.6	1900 mL/day ^#^	2000 mL/day ^#^	
**Micronutrients**															
Vitamin D			100.0	0.0			100.0	0.0			100.0	0.0	20 μg/day		
Folate			34.4	65.6			73.8	26.2			69.6	30.4	240 μg/day	300 μg/day	
Vitamin B12			36.5	63.5			50.0	50.0			52.2	47.8	3.5 μg/day	4 μg/day	
Fe			51.0	49.0			54.8	45.2			66.3	33.7	15 mg/day		
Mg			20.8	79.2			45.2	54.8			62.4	32.6	250 mg/day	310 mg/day	350 mg/day
Ca			79.2	20.8			85.7	14.3			82.6	17.4	1100 mg/day	1200 mg/day	
P			37.5	62.5			33.3	66.7			50.0	50.0	1250 mg/day		

Note: recommendations for nutrients and dietary fibre according to D-A-CH nutrient reference values (DRVs) [38]. ^#^ Recommendations for total water according to adequate intake in EFSA’s dietary reference values for water [36]. ^a^ WHO guidelines for free sugar intake [39]. ^b^ Total water intake refers to the total water from beverages and solid foods.

**Table 9 nutrients-16-01912-t009:** The proportion (%) of the male SI.Menu study population meeting daily dietary reference values for intake of macronutrients and micronutrients.

Macronutrients	Male												Dietary Reference Values (DRVs)		
	10–12 Years				13–14 Years				15–17 Years						
	% Meet DRI	% Do Not Meet DRI	% Below DRI	% Above DRI	% Meet DRI	% Do Not Meet DRI	% Below DRI	% Above DRI	% Meet DRI	% Do Not Meet DRI	% Below DRI	% Above DRI	10–12 Years	13–14 Years	15–17 Years
Carbohydrates	40.8	59.2			39.2	60.8			33.7	66.3			>50% of total energy intake		
Free sugars (criteria 1) ^a^	51.0	49.0			49.0	51.0			50.6	49.4			<10% of total energy intake		
Free sugars (criteria 2) ^a^	16.3	83.7			17.6	82.4			13.5	86.5			<5% of total energy intake		
Proteins (criteria 1)			2.0	98.0			3.4	96.6			6.7	93.3	42 g/day	57 g/day	66 g/day
Proteins (criteria 2)			1.0	99.9			5.9	94.1			11.2	88.8	0.9 g/kg of b. w.		0.9 g/kg of b. w.
Total fats	12.2	87.8	83.7	4.1	15.7	84.3	84.3	0.0			75.3	24.7	30–35% of TEI		30% of TEI
Dietary fibre			72.4	27.6			58.8	41.2			82.0	18.0	at least 19 g/day		at least 21 g/day
Total water ^b^			62.2	37.8			76.5	23.5			56.2	43.8	2100 mL/day ^#^	2500 mL/day ^#^	
**Micronutrients**															
Vitamin D			100.0	0.0			100.0	0.0			100.0	0.0	20 μg/day		
Folate			32.7	67.3			52.9	47.1			58.4	41.6	240 μg/day	300 μg/day	
Vitamin B12			21.4	78.6			17.6	82.4			22.5	77.5	3.5 μg/day	4 μg/day	
Fe			15.3	84.7			2.0	98.0			12.4	87.6	12 mg/day		
Mg			6.1	93.9			37.8	62.2			67.3	32.7	230 mg/day	310 mg/day	400 mg/day
Ca			59.2	40.8			68.6	31.4			61.8	38.2	1100 mg/day	1200 mg/day	
P			23.5	76.5			7.8	92.2			21.3	78.7	1250 mg/day		

Note: recommendations for nutrients and dietary fibre according to D-A-CH nutrient reference values (DRVs) [15]. ^#^ Recommendations for total water according to adequate intake in EFSA’s dietary reference values for water [33]. ^a^ WHO guidelines for free sugar intake [35]. ^b^ Total water intake refers to the total water from beverages and solid foods.

## Data Availability

Food consumption data presented in this study are available on the EFSA web page: https://www.efsa.europa.eu/en/data-report/food-consumption-data (accessed on 28 January 2022); other and more detailed data are available on request from the National Institute of Public Health, e-mail: EUmenu.SLO@nijz.si.

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
