# Peer review of "Dietary Intake of Adolescents and Alignment with Recommendations for Healthy and Sustainable Diets: Results of the SI.Menu Study"

_nutrients, 2024, doi:10.3390/nu16121912_

Round 1
Reviewer 1 Report
Comments and Suggestions for Authors
The researcher Rok Poličnik and collaborators propose a work that aims to assess dietary intakes (energy and nutrient intakes and food consumption) of Slovenian adolescents aged 10-17 years and to compare the intakes with dietary recommendations for healthy and sustainable diets. Some of the improvements that could be implemented are shown below:
- I consider that the summary could be shortened, it is too long when in fact there are data that will be provided later more concretely.
- Bibliographic contributions from 2005 related to BMI or 2009 related to menus could surely be updated, 19 years have passed.
- In line 71, reference number 16 is repeated.
- It is possible that the last paragraph of the introduction could be improved by simply stating objectives and hypotheses of results rather than positioning or emphasizing the situation of nutritional habits in Slovenia, which can be done previously.
- Sections on methodologies and results are quite explicit.
- Consider creating a new section that separates the discussion from the practical applications as well as the lines of research to give more importance to them.
I would like the authors to justify the use of these data as it is possible, perhaps even due to the influence of the pandemic, that habits among adolescents have changed, so I do not know if these results are as relevant today as they believe they are.
Author Response
Dear Sir/Madam,
We are sending you our responses to your review. In the attachment, we have included the latest version of the revised article and supplementary tables, which contains your suggestions and those of the other reviewers.
*************
- Summary
Thank you very much for taking the time to review this manuscript. Please find the detailed responses below and the corresponding revisions in track changes in the re-submitted files.
- Point-by-point response to Comments and Suggestions for Authors
Comments 1: The researcher Rok Poličnik and collaborators propose a work that aims to assess dietary intakes (energy and nutrient intakes and food consumption) of Slovenian adolescents aged 10-17 years and to compare the intakes with dietary recommendations for healthy and sustainable diets. Some of the improvements that could be implemented are shown below:
I consider that the summary could be shortened, it is too long when in fact there are data that will be provided later more concretely.
Response 1: Thank you for pointing this out. We agree with this comment. Therefore, we have shortened the abstract. We attach the revised article, where your comments are addressed in track changes.
Comments 2: Bibliographic contributions from 2005 related to BMI or 2009 related to menus could surely be updated, 19 years have passed.
Response 2: Agree. We have updated the reference. Your comment is addressed in in track changes.
Comments 3: In line 71, reference number 16 is repeated.
Response 3: Thank you for the correction. We have removed the duplicated reference from the text.
Comments 4: It is possible that the last paragraph of the introduction could be improved by simply stating objectives and hypotheses of results rather than positioning or emphasizing the situation of nutritional habits in Slovenia, which can be done previously.
Response 4: Thank you for pointing this out. We agree with your suggestion. We have revised the text of the last paragraph in the introduction chapter.
We attach the revised article, where your comments are addressed in track changes.
Comments 5: Consider creating a new section that separates the discussion from the practical applications as well as the lines of research to give more importance to them.
Response 5: Thank you for your suggestion. We have partially incorporated it in the manuscript by including additional text in the discussion chapter (line 543-550) that illustrates the importance of the findings for practical use. In the future, it will be necessary to promote the importance of sustainable aspects in dietary habits among adolescents and other population groups. We attach the revised manuscript, where your comments are addressed in track changes.
Comments 6: I would like the authors to justify the use of these data as it is possible, perhaps even due to the influence of the pandemic, that habits among adolescents have changed, so I do not know if these results are as relevant today as they believe they are.
Response 6: The study on the dietary intake of adolescents and other population groups in Slovenia was conducted before the COVID-19 pandemic. The impact of the pandemic on the diet of adolescents was therefore not considered in our manuscript. A new national survey on dietary intake is planned, which may reveal any changes in eating habits after the pandemic period.
Kind regards,
Authors

Reviewer 2 Report
Comments and Suggestions for Authors
This is an interesting study describing the dietary intake of Slovenian adolescents aged 10-17 years.
Although the aims of the study may be of interest to the scientific community, the article has important limitations which, in my view, should be reviewed before publication.
I have added my contributions in the form of comments in the attached document.
Kind regards

Author Response
Dear Sir/Madam,
Thank you very much for your review of the article titled: "Dietary intake of adolescents and alignment with recommendations for healthy and sustainable diets: results of the SI.Menu study." We have responded to your comments in the attached document, and your suggestions have been incorporated into the revised version of the article.
Unfortunately, we were unable to attach the revised version of the article with visible corrections in track changes.
Best regards,
Authors

Reviewer 3 Report
Comments and Suggestions for Authors
Manuscript ID: nutrients-3025903
Type of manuscript: Article
Title: Dietary intake of adolescents and alignment with recommendations for healthy and sustainable diets: results of the SI.Menu study
A very well thought out and written work. Design and analysis is very important and attention should be paid to the future behavior of young people.
However, I have a few questions and comments that are worth taking into account in the work:
This point may be particularly interesting in the context of providing meals to poorer students.
- very well described in accordance with the EFSA standard, recommended study protocol, direct interviews and 122 anthropometric measurements, taking into account participants' height and weight
- Is health education currently carried out among students/teachers/employees in Slovenia?
- are there preventive programs regarding healthy eating in Slovenia? Are they possibly implemented as part of primary education in schools?
- I suggest that in Table 6 and 7 also provide kcal/day and not only kJ/day, since the authors refer the results in the Discussion to the energy value to ''other European countries [39–42].'' - line 339
- I have no major comments apart from these small changes in Table 6 and Table 7 because most similar studies (regardless of whether they are analyzes of adults or children, are given mainly in kcal units)
Author Response
Response to Reviewer 3 Comments
Title of manuscript: Dietary intake of adolescents and alignment with recommendations for healthy and sustainable diets: results of the SI.Menu study
Dear Sir/Madam,
We are sending you our responses to your review. In the attachment, we have included the latest version of the revised article, which contains your suggestions and those of the other reviewers.
******
- Summary
Thank you very much for taking the time to review this manuscript. Please find the detailed responses below and the corresponding revisions highlighted in the re-submitted files.
- Questions for General Evaluation Reviewer’s Evaluation Response and Revisions
Does the introduction provide sufficient background and include all relevant references? Yes /
Are all the cited references relevant to the research? Yes /
Is the research design appropriate? Yes /
Are the methods adequately described? Yes /
Are the results clearly presented? Yes /
Are the conclusions supported by the results? Yes /
- Point-by-point response to Comments and Suggestions for Authors
A very well thought out and written work. Design and analysis is very important and attention should be paid to the future behavior of young people. However, I have a few questions and comments that are worth taking into account in the work: This point may be particularly interesting in the context of providing meals to poorer students
Comments 1: Very well described in accordance with the EFSA standard, recommended study protocol, direct interviews and 122 anthropometric measurements, taking into account participants' height and weight
Response 1: Thank you very much for you comment.
Comments 2: Is health education currently carried out among students/teachers/employees in Slovenia?
Response 2: Thank you for your question. The health education program in Slovenia is intended for all children and adolescents from birth to the end of schooling, as well as for families, parents, students, other young people, educators, and others. It is linked with programs designed for pregnant women, future parents, and parents of infants. Activities take place in health center facilities, educational institutions, and the local community. The program is conducted by healthcare professionals, mostly registered nurses, as well as doctors, psychologists, physiotherapists, kinesiologists, certified midwives, and others. The health education program for children and adolescents is conducted at the primary level of healthcare and is part of the healthcare system. The field is regulated by a decree issued by the Ministry of Health.
Comments 3: Are there preventive programs regarding healthy eating in Slovenia? Are they possibly implemented as part of primary education in schools?
Response 3: Similar to several European countries, Slovenia has a longstanding tradition of school meal systems, dating back to 1949. Governed by the School Meals Act, this legislation serves as the foundation for organizing, subsidizing, and ensuring meal quality in educational institutions. Additionally, it mandates the education of adolescents on healthy diets. The law also addresses the needs of children from lower socio-economic backgrounds, offering subsidies for individual students. Practical guidelines for school menu planning were introduced four decades ago, followed by national dietary guidelines for educational institutions in 2005. In 2023, new dietary recommendations were adopted for the school meal program. Primary schools are required to provide at least one daily meal, typically a morning snack, with additional meals available at parents' expense. Parents cover the cost of food, while national and local governments handle overhead expenses for school kitchen operations. The nutritional quality of school meals is monitored by the National Institute of Public Health.
Comments 4: I suggest that in Table 6 and 7 also provide kcal/day and not only kJ/day, since the authors refer the results in the Discussion to the energy value to ''other European countries [39–42].'' - line 339
Response 4: Thank you for your suggestion. We agree with you that articles mention various units of measurement for the energy value of food (kcal, kJ, MJ, etc.). However, as authors, we have decided to present our data in kJ, which is also the official unit of measurement. One of the arguments for why we decided to cite data only in kJ is that we used the same citation method in previous articles, which included data from the Si Menu study.
Comments 5: I have no major comments apart from these small changes in Table 6 and Table 7 because most similar studies (regardless of whether they are analyzes of adults or children, are given mainly in kcal units)
Response 5: Thank you once again for your comments and suggestions for improving the article. We would like to add that in previous manuscripts based on the Si Menu study, reviewers commented that we must use the unit kJ or MJ. Due to numerous past comments, we have also used only kJ for citing data in this article.
Kind regards,
Authors

Round 2
Reviewer 2 Report
Comments and Suggestions for Authors
Although the authors have done a very good job of revising the article, incorporating the contributions made in a very good way, there is one key issue that remains unresolved:
- It is not clear how the influence of different socio-demographic factors on food intake has been assessed (Supplementary Table 3). What kind of statistical analysis has been performed and what are the results of this statistical analysis to consider that these determinants are related to dietary intake?
- This information is not included in the results of the study. If it is one of the objectives of the study, it should be included in the results section and not only referred to in the discussion section.
Apart from this key aspect, there are other small aspects that could be revised:
- In the objective, instead of saying that they assess the influence of variables such as age and sex on the intake of different foods, they could state that they assess the influence of different socio-demographic variables.
- They should clearly explain in the methodology why in the analysis by age groups they sometimes use two groups and sometimes three.
- Do all adolescents in Slovenia eat at school? If not, the influence of eating at school could have been assessed...
Best regards
Author Response
|
Response to Reviewer 2 Comments |
Title of manuscript: Dietary intake of adolescents and alignment with recommendations for healthy and sustainable diets: results of the SI.Menu study
|
Thank you very much for taking the time to review this manuscript. Please find the detailed responses below and the corresponding revisions highlighted in the re-submitted files. |
|
|
2. Point-by-point response to Comments and Suggestions for Authors |
|
|
Comments 1: Although the authors have done a very good job of revising the article, incorporating the contributions made in a very good way, there is one key issue that remains unresolved: It is not clear how the influence of different socio-demographic factors on food intake has been assessed (Supplementary Table 3). What kind of statistical analysis has been performed and what are the results of this statistical analysis to consider that these determinants are related to dietary intake? Response 1: We appreciate your insightful suggestion regarding the explanation of LASSO method within the statistical section. Balancing conciseness and detailed explanations are crucial. Upon your recommendation to focus on the link between LASSO and the analysis outcomes, we made improvements in the text also acknowledging that detailed LASSO methodology can be found in the referenced paper. We've implemented this approach, and we hope that this time, we ensure a clear understanding of LASSO's role in our analysis. We believe this revision strikes a good balance between clarity and comprehensiveness for journal scientific readership. Thank you for helping us enhance this section and improving our manuscript. Comments 2: This information is not included in the results of the study. If it is one of the objectives of the study, it should be included in the results section and not only referred to in the discussion section. Response 2: We appreciate your suggestion to present the results from Supplementary Table 3 in the results section. We've addressed this by adding a new paragraph within the results section that summarizes the key findings. Additionally, we've included a clear reference to the supplementary file, directing readers there for more detailed information. This approach ensures a concise presentation of the main results within the body of the paper, while providing access to the full data for those who require it. Comments 3: Apart from this key aspect, there are other small aspects that could be revised: In the objective, instead of saying that they assess the influence of variables such as age and sex on the intake of different foods, they could state that they assess the influence of different socio-demographic variables. Response 3: Thank you for the recommendation, which we have taken into account in the last sentence of the introduction section.
Comments 4: They should clearly explain in the methodology why in the analysis by age groups they sometimes use two groups and sometimes three. Response 4: Thank you for pointing out this issue. In section 2.1. Study design and population, we have added two sentences explaining why the adolescents are divided into three or two groups. |
|
|
Comments 5: Do all adolescents in Slovenia eat at school? If not, the influence of eating at school could have been assessed... Response 5: Thank you for asking this question. To clarify, Slovenia has a longstanding tradition of school meal systems. Governed by the School Meals Act from 2010, this legislation serves as the foundation for organizing, subsidizing, and ensuring meal quality in educational institutions. Additionally, it mandates the education of adolescents on healthy diets. The law also addresses the needs of children from lower socio-economic backgrounds, offering subsidies for individual students. In 2023, new dietary recommendations were adopted for the school meal program. Primary schools are required to provide at least one daily meal, typically a morning snack, with additional meals available at parents' expense. The nutritional quality of school meals is monitored by the National Institute of Public Health. Dietary data collection employed two 24-hour dietary recalls (24HRs) and a Food Frequency Questionnaire (FPQ), following the European Food Safety Authority (EFSA) methodology. The 24HRs were conducted on non-consecutive days and across different seasons to capture potential variations in dietary intake. All food items consumed by participants during the previous day were recorded. This comprehensive approach provides a detailed picture of dietary intake. Although, meals provided at school and at home were included, they were not specifically coded in order to be separately analyzed and their influence in total diet provided. We appreciate your suggestion, and this could be taken into consideration in other project that will deal with the influence of school meals on adolescent total diets. To address potential ambiguity regarding national regulations for school meals in the primary education sector, we have incorporated an additional sentence within the discussion section. This sentence clarifies that, by law, all primary schools are mandated to provide one daily meal to students. The option exists for schools to offer additional meals if necessary. |
Kind regards,
Authors
